# Short dual antiplatelet therapy duration after percutaneous coronary intervention in high bleeding risk patients: Systematic review and meta-analysis

**Kevin R. Bainey[1], Guillaume Marquis-Gravel[2], Blair J. MacDonald[3], David Bewick[4], Andrew Yan[5], Ricky D. Turgeon[3]\***

1 Mazankowski Alberta Heart Institute, University of Alberta, Edmonton, Alberta, 2 Montreal Heart Institute, Université de Montréal, Montreal, Quebec, 3 Faculty of Pharmaceutical Sciences, University of British Columbia, Vancouver, British Columbia, 4 New Brunswick Heart Center, Horizon Health Network, Saint John, New Brunswick, 5 Division of Cardiology, Canadian Heart Research Centre and Terrence Donnelly Heart Centre, St Michael's Hospital, University of Toronto, Toronto, Ontario

\* ricky.turgeon@ubc.ca

**Data Availability Statement:** All relevant data are within the paper and its Supporting Information files.

## Abstract

### Introduction

Dual antiplatelet therapy (DAPT) following percutaneous coronary intervention (PCI) reduces major adverse cardiovascular events (MACE) and stent thrombosis. However, DAPT duration is a concern in high bleeding risk (HBR) patients. We evaluated the effect of short DAPT (1–3 months) compared to standard DAPT (6–12 months) on bleeding and ischemic events in HBR PCI.

### Methods

We searched MEDLINE, Embase and CENTRAL up to August 18, 2022. Randomized controlled trials (RCTs) comparing short DAPT (1–3 months) versus standard DAPT in HBR PCI were included. We assessed risk of bias (RoB) using the Cochrane RoB2 tool, and certainty of evidence using GRADE criteria. Outcomes included MACE, all-cause death, stent thrombosis, major bleeding, and the composite of major or clinically-relevant non-major bleeding. We estimated risk ratios (RR) and 95% confidence intervals (CI) using a random-effects model.

### Results

From 503 articles, we included five RCTs (n = 7,242) at overall low risk of bias with median follow-up of 12-months. Compared to standard DAPT, short DAPT did not increase MACE (RR 1.02, 95% CI 0.84–1.23), all-cause death (RR 0.92, 95% CI 0.71–1.20) or stent thrombosis (RR 1.47, 95% CI 0.73–2.93). Short DAPT reduced major bleeding (RR 0.34, 95% CI 0.13–0.90) and the composite of major or clinically-relevant non-major bleeding (RR 0.60, 95% CI 0.44–0.81), translating to 21 and 34 fewer events, respectively, per 1000 patients.

**Funding:** This work was conducted to support the Canadian Cardiovascular Society Antiplatelet Guidelines with funding provided to Ricky Turgeon by the Canadian Cardiovascular Society (https://ccs.ca/). The funder did not play any role in the study design, data collection or analysis, decision to publish, or preparation of the manuscript.

**Competing interests:** The authors have declared that no competing interests exist.

## Conclusions

In HBR PCI, DAPT for 1–3 months compared to 6–12 months reduced clinically-relevant bleeding events without jeopardizing ischemic risk. Short DAPT should be considered in HBR patients receiving PCI.

## Introduction

Dual antiplatelet therapy (DAPT) following percutaneous coronary intervention (PCI) reduces the risk of stent thrombosis and protects from non-culprit atherothrombotic events long-term. The 2018 Canadian Cardiovascular Society (CCS)/Canadian Association of Interventional Cardiology (CAIC) Focused Update for the Use of Antiplatelet Therapy recommend DAPT with low-dose aspirin and a P2Y12 receptor inhibitor for 1 year (and up to 3 years in low bleeding / high ischemic risk patients) in acute coronary syndromes (ACS) and a minimum of 6 months (and up to 1 year) in non-ACS patients following PCI [1]. Subsequent studies in the era of use of newer-generation drug-eluting stents (DES), which have reduced risk of stent thrombosis, have suggested that DAPT may be safely be reduced to 6 months without increasing thrombotic risk [2–4]. More recently, this threshold has been further reduced with randomized studies supporting shortened DAPT of 1 to 3 months following PCI regardless of ACS presentation using mainly $P2Y_{12}$ inhibitor single antiplatelet therapy (SAPT) [5–10]. Standard (and prolonged) DAPT protects against thrombotic events, yet major bleed following PCI is associated with a 3 to 5-fold risk of mortality, which offsets this protective benefit [11]. In this context, identifying patients at high bleeding risk (HBR) may help tailor DAPT duration recommendations [12]. The Academic Research Consortium (ARC) have created a definition for HBR that includes major a minor risk criteria, with patients being classified as HBR if they fulfill at least 1 major or 2 minor criteria [13]. This HBR definition is associated with a 1-year risk of a Bleeding Academic Research Consortium (BARC) category 3 or 5 bleed $\geq$4% or intracranial hemorrhage (ICH) of $\geq$1%.

We performed a systematic review and meta-analysis of randomized studies in HBR patients receiving PCI with DES to determine whether short DAPT duration of 1 to 3 months followed by SAPT compared to standard DAPT duration of $\geq$6 months reduces the risk of major bleeding without compromising ischemic events.

## Methods

We performed a systematic review and meta-analysis according to the methodology outlined in the Cochrane Handbook for Systematic Reviews and Interventions using a prospectively-designed (but not registered) protocol, and reported following the 2020 PRISMA statement [11].

### Search strategy

We searched MEDLINE (inception to August 18, 2022), Embase (January 1, 2019 to August 18, 2022), and the Cochrane Central Register of Controlled Trials (CENTRAL; January 1, 2019 to August 18, 2022) using the search strategies described in the **S1 Appendix**. We further supplemented this with a Web of Science Reverse Citation Search of the initial ARC-HBR criteria article [13].

## Study selection and data extraction

We included parallel randomized controlled trials (RCTs) that enrolled patients who underwent PCI for either ACS or non-ACS indications and had HBR, or reported on a subgroup of patients with HBR, and compared DAPT for 1–3 months followed by SAPT ("short DAPT"), to DAPT for 6–12 months ("standard DAPT"). We included RCTs if they defined HBR based on the ARC-HBR criteria or other explicitly defined criteria.

One reviewer (RDT) performed all database searches and imported the records into Covidence. Using Covidence, two reviewers (BJM and RDT) independently screened article titles and abstracts, and reviewed full-text articles for inclusion. The same two reviewers independently extracted the following data from each study using a standardized data collection form: Study acronym, lead author, publication year, sample size (with HBR), definition of HBR used in the trial, proportion of patients meeting individual components of HBR definition, baseline characteristics (age, sex, percentage with ACS as indication for PCI), stent characteristics, DAPT duration in both groups, proportion receiving each P2Y12 inhibitor in DAPT regimen, antiplatelet continued as monotherapy, proportion receiving oral anticoagulant as co-intervention, follow-up duration, and data on all outcomes (number of participants with events and total number of participants in each group in the intention-to-treat population).

## Assessment of risk of bias and certainty of evidence

Two reviewers (BJM and RDT) independently evaluated trial-level risk of bias (RoB) using the Cochrane RoB 2 tool [14]. The same two reviewers then rated outcome-level certainty of evidence using the Grading Recommendations Assessment, Development and Evaluation (GRADE) framework, which incorporates risk of bias, imprecision, inconsistency, indirectness, and publication bias [15]. We planned to assess for publication bias by visual inspection of funnel plots for analyses with at least 10 trials because this test is considered to be of little value with fewer trials [11].

**Outcomes.** The outcomes of interest were: (1) major adverse cardiovascular events (MACE) (composite of death from any cause, myocardial infarction (MI), or stroke—when this composite was not available, we used the composite of cardiovascular death, MI, or stroke, or a broader composite that encompassed additional components); (2) all-cause death; (3) stent thrombosis (definite/probable based on original ARC [16] or ARC-2 [17] definition); (4) major bleed (BARC [18] classification [3 or 5 bleed] prioritized due to widespread use, and standardized, inclusive definitions; when information using this classification was not available, we preferentially extracted bleeding based on the International Society on Thrombosis and Haemostasis (ISTH), followed by the Thrombolysis in Myocardial Infarction (TIMI) definition; and (5) Major or non-major clinically-relevant bleeding (BARC [18] 2, 3, or 5 bleed).

## Statistical analysis

We pooled dichotomous outcomes as risk ratios (RR) with 95% confidence intervals (CIs) using a DerSimonian-Laird inverse variance random-effects model for all outcomes. We further presented absolute differences using simple frequencies (per 1000 patients treated). We evaluated statistical heterogeneity with visual inspection of the forest plot and quantified the percentage of the variability that is due to heterogeneity between trials using the $I^2$ statistic. Where possible, we performed pre-specified subgroup analyses based on PCI indication (ACS or non-ACS), concomitant use of oral anticoagulants (OAC), and choice of antiplatelet SAPT type following DAPT in the intervention arm. We conducted all analyses using Review Manager version 5.4 (Cochrane, Copenhagen, Denmark) and GRADEPro.org.

## Results

### Search and selection of studies

From 503 articles, we included five RCTs reported across 9 articles (**Fig 1**) involving 7,242 patients. The included studies (and subgroup analyses) were: MASTER DAPT (the only RCT exclusively enrolling patients with HBR) [19–22], and the HBR subgroup of TWILIGHT [23], TICO [24], and STOPDAPT-2 Total Cohort (which consisted of two trials: STOPDAPT-2 and STOPDAPT-2 ACS) [25–27]. **S1 Table** outlines the HBR definition used in each trial and proportion of patients fitting each criterion when reported. Across all five RCTs, mean age was 74 years, 30% were female, and 57% presented with ACS. Trial and patient characteristics are summarized in **Table 1**. Duration of DAPT prior to switch to SAPT was 1 month in MASTER DAPT, 1–2 months in STOPDAPT-2 and STOPDAPT-2 ACS, and 3 months in TWILIGHT and TICO. Post-DAPT SAPT consisted of P2Y$_{12}$ inhibitor monotherapy in all trials except MASTER DAPT, in which 28.8% received acetylsalicylic acid (ASA). Post-randomization follow-up was 11 months in MASTER DAPT and 12 months in all other trials (median 12-month follow-up). Overall RoB was low in four trials and uncertain in one trial (TICO-HBR; **S1 Fig**). We rated the certainty of evidence as high for major/clinically-relevant non-major bleed, moderate due to imprecision for MACE, death and stent thrombosis, and moderate due to inconsistency for major bleed (**Table 2**).

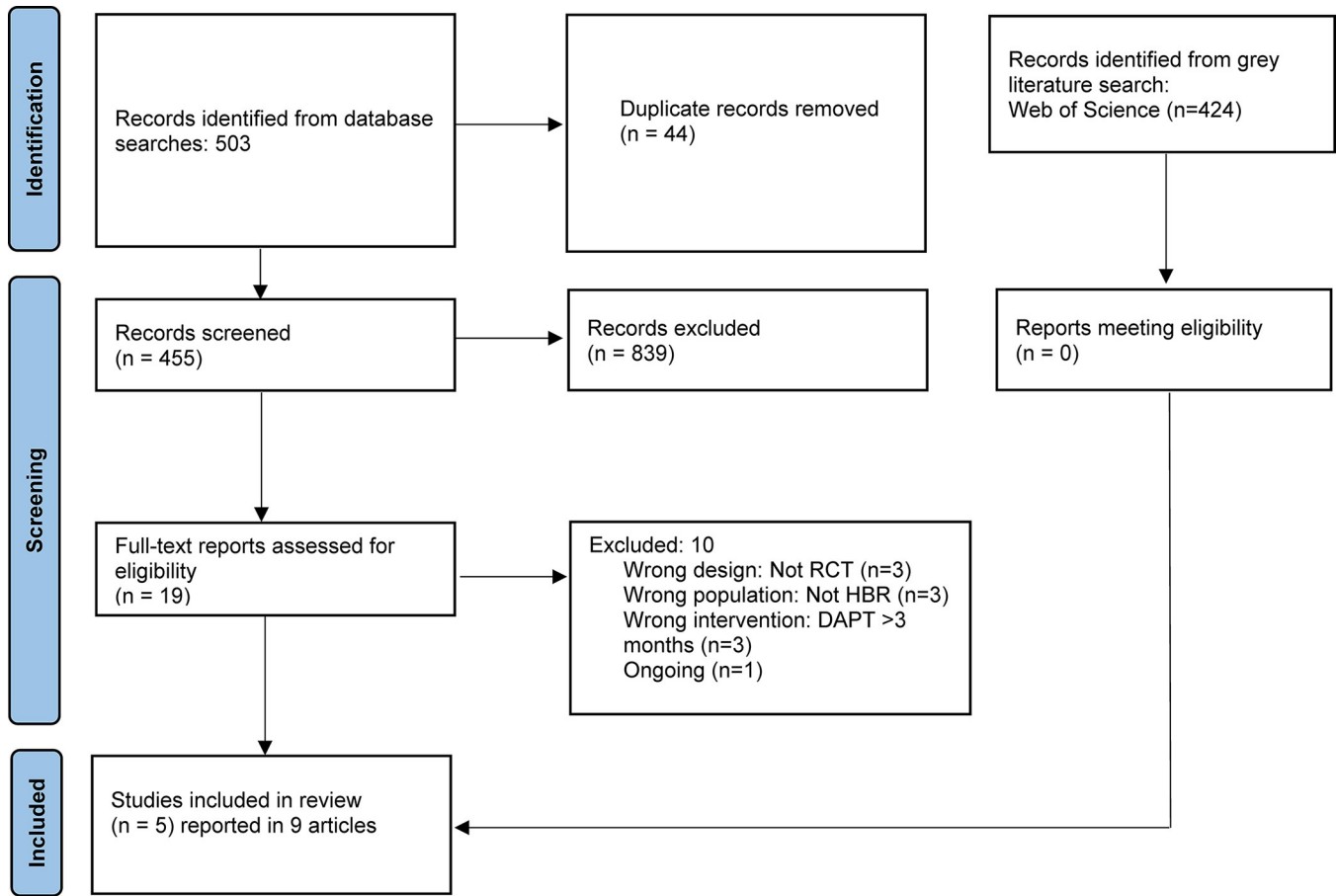

**Fig 1. Preferred Reporting Items for Systematic Reviews and Meta-Analyses (PRISMA) flow diagram.**

**Table 1. Study and patient characteristics.**

| | MASTER DAPT [19] | TWILIGHT-HBR [23] | TICO-HBR [24] | STOPDAPT-2 Total Cohort-HBR [27]* | STOPDAPT-2-HBR [25, 26]* |
|---|---|---|---|---|---|
| n | 4579 | 1064 | 453 | 1146 | 1054 |
| Mean age | 76 | 72 | 71 | 68 | 76 |
| Female sex, % | 31 | 33 | 36 | 22 | 30 |
| ACS as index event, % | 48 | 62 | 100 | 69 | 29 |
| Receiving OAC, % | 36 | 0 | | 1 | |
| Stent characteristics | Biodegradable polymer-coated sirolimus-eluting stent (Ultimaster) | Drug-eluting stents (97.8% second-generation) | Ultrathin bioresorbable polymer sirolimus-eluting stents | Cobalt-chromium everolimus-eluting stent | |
| **Intervention:** | | | | | |
| DAPT duration | 1 month | 3 months | | 1–2 months | |
| P2Y12i used in DAPT | Clopidogrel, prasugrel, or ticagrelor | Ticagrelor | | Clopidogrel | |
| Agent after DAPT | Agent used in SAPT 1 month after randomization: Clopidogrel (53.9%), aspirin (28.8%), ticagrelor (13.6%), prasugrel (1.2%) | Ticagrelor | | Clopidogrel | |
| **Comparator:** | | | | | |
| DAPT duration | Median 6.4 months | 12 months | 12 months | 12 months | 12 months |
| P2Y12i used | Clopidogrel, prasugrel, or ticagrelor | Ticagrelor | Ticagrelor | Clopidogrel | Clopidogrel |
| MACE definition | Death, MI, stroke | | Death, MI, stroke, stent thrombosis, target vessel revascularization | Cardiovascular death, MI, stroke, definite stent thrombosis | - |
| Major bleed definition | BARC 3 or 5 | | TIMI major | - | BARC 3 or 5 |
| Non-major, clinically-relevant bleed definition | BARC 2 | | - | TIMI minor | - |
| Timing of randomization | 1 month after PCI | 3 months after PCI | Index PCI | | |
| Follow-up duration after randomization | 11 months | 12 months | 12 months | 12 months | 12 months |

ACS, acute coronary syndrome; BARC, bleeding academic research consortium; DAPT, dual antiplatelet therapy; HBR, high bleeding risk; MACE, major adverse cardiovascular event; OAC, oral anticoagulation; PCI, percutaneous coronary intervention; SAPT, single antiplatelet therapy.

*STOPDAPT-2 Total Cohort-HBR included pooled HBR subgroup data from two distinct trials: STOPDAPT-2 and STOPDAPT-2 ACS, whereas STOPDAPT-2-HBR included HBR subgroup data for STOPDAPT-2 only (i.e. not STOPDAPT-2 ACS).

## Major adverse cardiovascular events, all-cause death, and stent thrombosis

All five trials reported MACE. There was no significant difference in MACE between short and standard-duration DAPT (RR 1.02, 95% CI 0.84–1.23, $I^2$ = 0%; Fig 2A, Table 2).

All trials except STOPDAPT-2 ACS reported on all-cause death and stent thrombosis. There were no significant differences between short and standard-duration DAPT for all-cause death (RR 0.92, 95% CI 0.71–1.20, $I^2$ = 0%; Fig 2B, Table 2) or stent thrombosis (RR 1.47, 95% CI 0.73–2.93, $I^2$ = 0%; Fig 2C, Table 2).

## Bleeding events

Short DAPT reduced the risk of major bleed compared with standard DAPT duration (RR 0.34, 95% CI 0.13–0.90, $I^2$ = 78%, Fig 3A, Table 2), translating to 21 fewer events per 1,000 HBR patients treated with short DAPT. Similarly, there was a reduction in major or clinically-relevant non-major bleed (RR 0.60, 95% CI 0.44–0.81, $I^2$ = 41% Fig 3B, Table 2), translating to 34 fewer events per 1,000 HBR patients treated with short DAPT. Statistical heterogeneity in

**Table 2. Summary of findings table.**

| Outcome | Certainty of evidence | Event rate per 1000 in standard DAPT group | Effect estimate | |
|---|---|---|---|---|
| | | | RR (95% CI) | Absolute change, per 1000 |
| MACE | Moderate* | 57 | 1.02 (0.84–1.23) | (from 9 fewer to 13 more) |
| Death | Moderate* | 32 | 0.92 (0.71–1.20) | (from 9 fewer to 6 more) |
| Stent thrombosis | Moderate* | 4 | 1.47 (0.73–2.93) | (from 1 fewer to 7 more) |
| Major bleed | Moderate† | 32 | **0.34 (0.13–0.90)** | **21 fewer (from 3 to 28 fewer)** |
| Major or clinically-relevant non-major bleed | High | 86 | **0.60 (0.44–0.81)** | **34 fewer (from 16 to 48 fewer)** |

CI: Confidence interval, DAPT: Dual antiplatelet therapy, RR: Risk ratio

* Rated down 1 category for serious imprecision.

† Rated down 1 category for serious inconsistency.

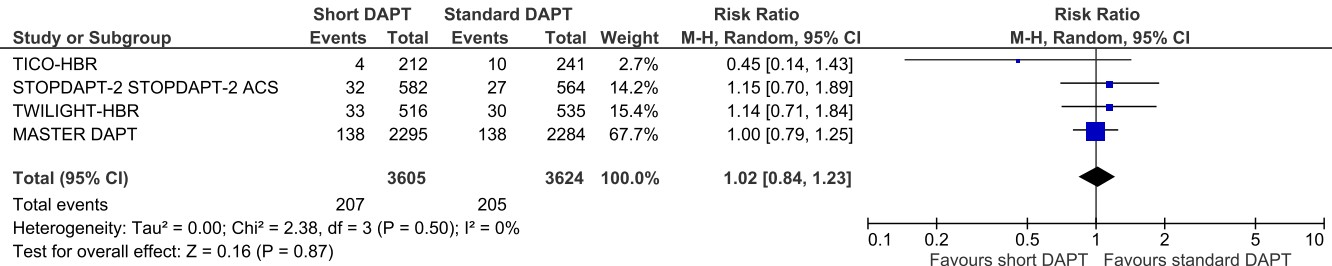

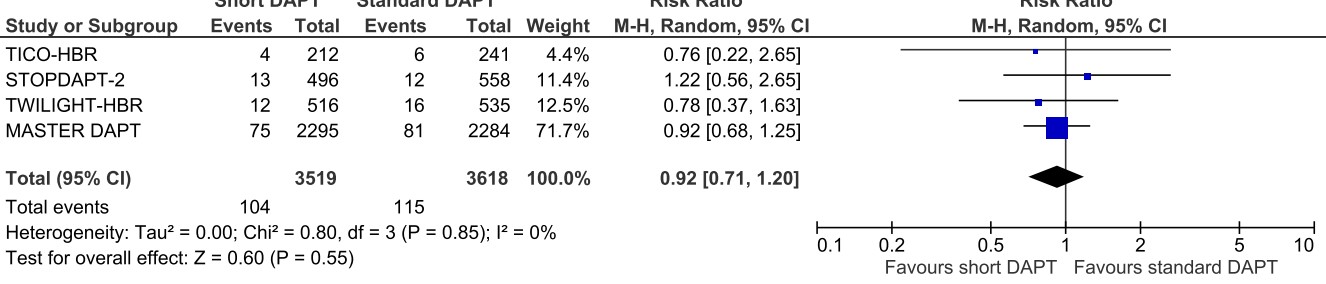

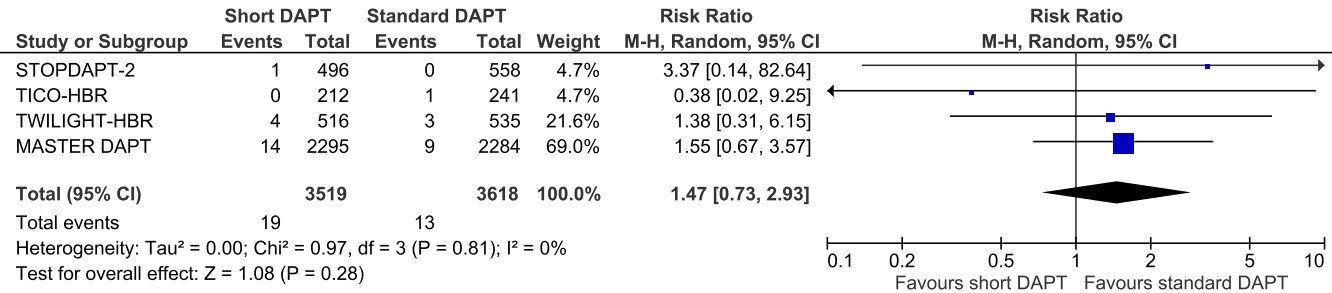

**Fig 2.** Forest plot of major adverse cardiovascular events (A), all-cause mortality (B) and stent thrombosis (C) with short dual antiplatelet therapy or standard dual antiplatelet therapy in patients with high bleeding risk receiving percutaneous coronary intervention. Squares and diamonds = risk ratios. Lines = 95% confidence intervals.

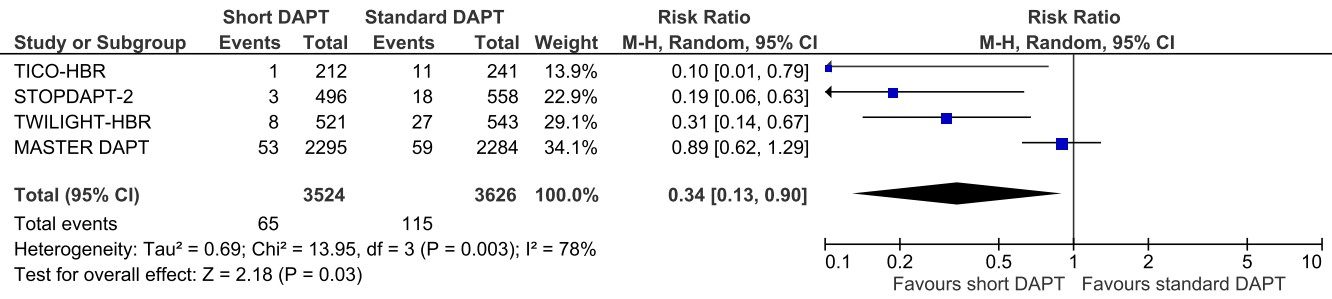

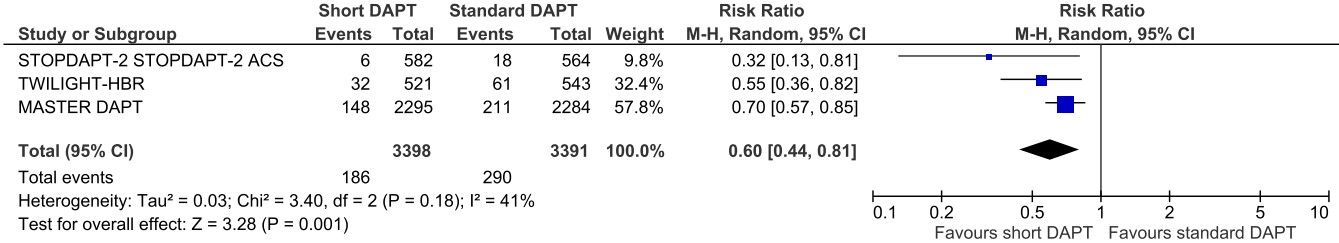

**Fig 3.** Forest plot of major bleeding (A) and major or clinically-relevant non-major bleeding (B) with short dual antiplatelet therapy or standard dual antiplatelet therapy in patients with high bleeding risk receiving percutaneous coronary intervention. Squares and diamonds = risk ratios. Lines = 95% confidence intervals.

both of these bleeding outcomes was mainly driven by quantitative and not directional differences between MASTER DAPT and the other trials.

## Subgroup analyses

The effect on all outcomes was consistent in subgroup analyses by ACS versus non-ACS (all p-interaction >0.10; **S2 Fig**). For the subgroup analysis based on receipt of OAC at baseline, there was a significant subgroup interaction for major or clinically-relevant non-major bleeding (p-interaction = 0.02); however, this included only two trials (and data from MASTER DAPT trial in both subgroups) and was not corroborated by a subgroup difference for major bleeding (p-interaction = 0.22; **S3 Fig**). For the subgroup analysis based on SAPT type after DAPT in the short DAPT arm, there was a significant subgroup interaction for major bleed (p-interaction = 0.002), though this was again driven by differences between MASTER DAPT and other trials (**S4 Fig**).

## Discussion

This comprehensive systematic review and meta-analysis of HBR PCI patients demonstrated a reduction in bleeding events with shortening to SAPT following 1 to 3 months of DAPT compared to a DAPT duration of 6–12 months, without compromising MACE or all-cause mortality (**Fig 4**). Furthermore, the risk of stent thrombosis was ≤0.5% regardless of DAPT duration. Finally, these results were consistent regardless of clinical presentation, concomitant indication for OAC, or choice of SAPT following DAPT, though most patients received P2Y$_{12}$ inhibitor monotherapy.

The present study extends the results of prior systematic reviews that have focused primarily on patients without HBR undergoing PCI [28–31]. Consistent with our results, these previous systematic reviews found no further reduction in thrombotic events with standard or prolonged DAPT (≥12 months) compared with an abbreviated course followed by SAPT.

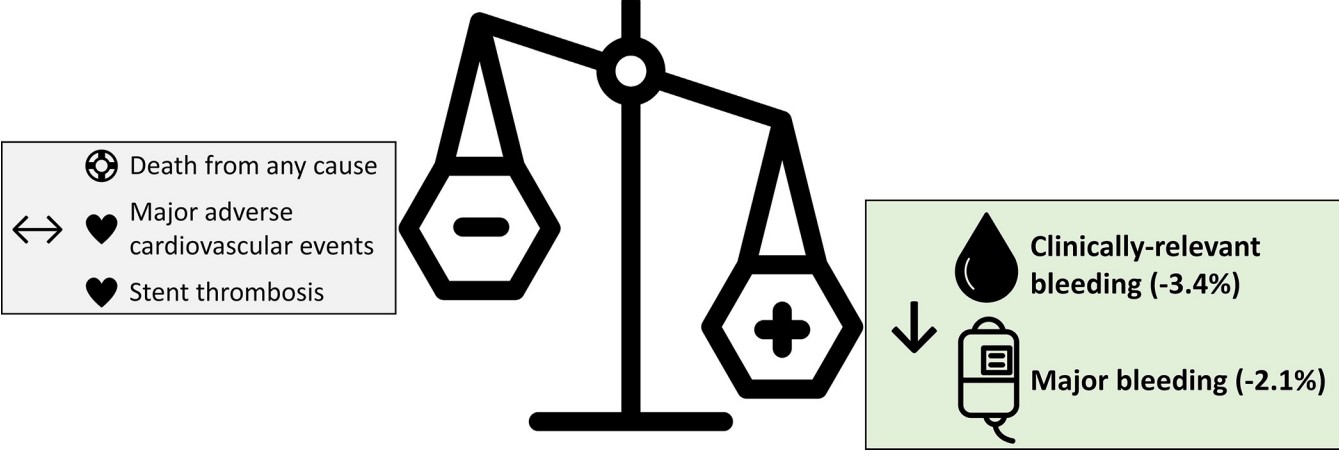

**Fig 4. Central illustration.** HBR: High bleeding risk; DAPT: Dual antiplatelet therapy.

Conversely, our study found a statistically significant and clinically meaningful reduction in major bleeding from reducing DAPT duration to 1–3 months in HBR patients, which has only inconsistently been observed in studies of non-HBR patients [28–31]. A recent meta-analysis of 9006 HBR patients–excluding patients with OAC–similarly found a reduction in bleeding events without compromising thrombotic outcomes [32]. Patients requiring an OAC have historically been excluded from prior DAPT studies, yet this is a key bleeding risk factor that impacts decisions around DAPT selection and duration [13]. The MASTER DAPT trial included in our review specifically included a subgroup of patients requiring OAC and stratified randomization according to indication for OAC [22]. In subgroup analyses within the present meta-analysis, bleeding reduction was greatest in patients without concomitant OAC; however, there was no increase in MACE with or without OAC indication.

Shortening DAPT duration to 1–3 months in HBR patients did not appear to compromise thrombotic risk, as there was no significant increase in MACE or all-cause death, and stent thrombosis risk remained below 0.5% in both groups over trial follow-up. Notably, prior studies have demonstrated that the greatest risk of stent thrombosis occurs in the first 30 days after stent placement and markedly attenuates thereafter, and therefore even short DAPT provides protection during the highest-risk period [33]. This universally low risk of stent thrombosis may have been driven by use of newer-generation stents in the included trials, which also mimics use in contemporary practice. MASTER DAPT and TICO used thin biodegradable-polymer DES [8, 19], while the STOPDAPT-2 trials used a cobalt chromium permanent polymer DES [27] and TWILIGHT used various second-generation DES [7]. Additional studies demonstrate that short DAPT appears safe even with complex PCI [21]. It remains unclear which antiplatelet agent would be best for SAPT following short DAPT. Most patients in the present study received a P2Y$_{12}$ inhibitor (clopidogrel or ticagrelor) following short DAPT. Additional studies suggest that a P2Y$_{12}$ inhibitor may be superior to acetylsalicylic acid monotherapy following DAPT. The HOST-EXAM trial conducted in South Korea found that clopidogrel 75 mg daily reduced the risk of MACE and major bleeding compared with ASA 100 mg daily after 6–18 months of DAPT in DES PCI [34]. Additionally, the CAPRIE trial had previously demonstrated lower risk of MI and bleeding with clopidogrel 75 mg daily compared with ASA 325 mg daily [35].

While the focus of our study was to evaluate the safety of short DAPT, other mitigating strategies to reduce bleeding risk have been studied. Most recently, DAPT de-escalation to less potent DAPT has garnered attention. Both TALOS-AMI and TOPIC addressed de-escalation of potent DAPT to clopidogrel-based DAPT 1-month following PCI for ACS. Both found a reduction in major bleeding events without increased risk in ischemic or stent thrombosis [36, 37]. The HOST-REDUCE-POLYTECH-ACS trial found comparable risk of bleeding and thrombotic events with prasugrel dose de-escalation versus a standard dose [38]. Overall, short DAPT has the strongest and most consistent evidence base, though head-to-head trials comparing these different strategies are needed.

Our study does come with limitations. First, we had access only to trial-level data. Second, most of the included studies were subgroup analyses of HBR patients from the randomized trial. Finally, the optimal choice of SAPT following short DAPT remains uncertain, though included studies predominantly stepped down to $P2Y_{12}$ inhibitor SAPT.

## Conclusion

This comprehensive systematic review with meta-analysis demonstrated reduced risk of bleeding events with short DAPT compared with standard DAPT duration in HBR patients undergoing PCI, with no significant increase in MACE, death, or stent thrombosis. This evidence should inform clinical decision-making about DAPT duration in this vulnerable population.

## Supporting information

**S1 Checklist. PRISMA 2020 checklist.**
(DOCX)

**S1 Appendix. Database search strategy.**
(DOCX)

**S1 Table. Definition of high bleeding risk and proportion fitting each criterion within each trial.**
(DOCX)

**S1 Fig. Risk of bias assessment.**
(DOCX)

**S2 Fig. Subgroup based on acute coronary syndrome (ACS) versus non-ACS.**
(DOCX)

**S3 Fig. Subgroup based on receipt of oral anticoagulant (OAC) at baseline.**
(DOCX)

**S4 Fig. Subgroup based on single antiplatelet therapy (SAPT) choice following dual antiplatelet therapy (DAPT) in the short DAPT arm.**
(DOCX)

## Author Contributions

**Data curation:** Ricky D. Turgeon.

**Formal analysis:** Blair J. MacDonald, Ricky D. Turgeon.

**Funding acquisition:** Ricky D. Turgeon.

**Investigation:** Ricky D. Turgeon.

**Methodology:** Ricky D. Turgeon.

**Project administration:** Ricky D. Turgeon.

**Supervision:** Ricky D. Turgeon.

**Visualization:** Ricky D. Turgeon.

**Writing – original draft:** Kevin R. Bainey.

**Writing – review & editing:** Guillaume Marquis-Gravel, Blair J. MacDonald, David Bewick, Andrew Yan, Ricky D. Turgeon.

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
