## [Decision Letter · Decision Letter 0]

25 Apr 2023

PONE-D-23-06603Short Dual Antiplatelet Therapy Duration After Percutaneous Coronary Intervention in High Bleeding Risk Patients: Systematic Review and Meta-analysisPLOS ONE

Dear Dr. Turgeon,

Thank you for submitting your manuscript to PLOS ONE. After careful consideration, we feel that it has merit but does not fully meet PLOS ONE’s publication criteria as it currently stands. Therefore, we invite you to submit a revised version of the manuscript that addresses the points raised during the review process.

Please address each reviewers' comment individually in the response letter indicating where in the manuscript you have provided revisions. 

We look forward to receiving your revised manuscript.

Kind regards,

R. Jay Widmer

Academic Editor

PLOS ONE

Additional Editor Comments:

In general, the reviewers were fairly positive regarding the paper and its primary finding/message. However, there are serious concerns regarding the methods, statistics, and protocols used in the meta-analysis and how those might apply to the discussion. Furthermore, improved graphics and figures could be incorporated. Please make every effort to utilize these comments to improve the manuscript as written.

Reviewers' comments:

Reviewer's Responses to Questions

**Comments to the Author**

1. Is the manuscript technically sound, and do the data support the conclusions?

Reviewer #1: Partly

Reviewer #2: Yes

2. Has the statistical analysis been performed appropriately and rigorously? 

Reviewer #1: Yes

Reviewer #2: No

3. Have the authors made all data underlying the findings in their manuscript fully available?

Reviewer #1: Yes

Reviewer #2: Yes

4. Is the manuscript presented in an intelligible fashion and written in standard English?

Reviewer #1: Yes

Reviewer #2: Yes

5. Review Comments to the Author

Reviewer #1: Dr. Bainey and co-authors present a meta-analysis of short vs. standard antiplatelet therapy in HBR patients. Inherent limitations is the study level design.

My comments are as follow:

- Figure S1 is not clear, there is some overlap between column 1 and 2. How can a RCT have higher risk of bias in domain 2 than post-hoc analyses? Please consider reviewing.

- Many studies on DAPT duration in HBR are non-randomized. I would understand their exclusion if you had included only RCT. However, this is not the case, as sub-analyses of previous RCT (possibly even non pre-specified) were included as well).

- You should provide definitions of bleeding and ischemic events. The fact that they are not consistent across studies needs to be taken into account.

- Publication bias should be assessed using funnel plots.

- Why did you favored BARC bleeding definition over TIMI? Please comment and discuss.

- I think a proper discussion on the literature is largely missing. There are several previous articles that should be cited and discussed in the paper. There is a wealth of papers on short DAPT, if one embark in another article in such topic, the discussion cannot ignore what has already been published, and most importantly should highlight why the work is different. Please refer to the following (which are a few but there are certainly more) and note that some of them are specific for patients at HBR:

1. doi: 10.1093/ehjcvp/pvaa127 - PMID: 33135064

2. doi: 10.1002/ccd.26110. Epub 2015 Aug 26 - PMID: 26309050 

3. doi: 10.1093/ehjcvp/pvac065. Online ahead of print - PMID: 36427063

4. doi: 10.1093/ehjcvp/pvaa001 - PMID: 31942965

5. doi: 10.1093/eurheartj/ehac706 - PMID: 36477292

- It would be really nice to summarize the main finding of the study in a central illustration. The paper lacks figures, which ensure clarity.

Reviewer #2: In this analysis, Bainey and colleagues evaluated the effect of short DAPT (1-3 months) compared to standard DAPT (6-12 months) on bleeding and ischemic events in HBR PCI patients.

After a careful evaluation of the manuscript, these are my concerns.

The authors declared that the methodology outlined in the Cochrane Handbook for Systematic Reviews and Interventions, and reported following the 2020 PRISMA statement. However, the authors did not provide, among the others, the registration information for the review, nor report the reasons for the missing items in the limitation section.

Furthermore, the exclusion criteria (type of the analyses exclude as language, abstract presentation to congresses, and so on) were not detailed in the method section.

Please report in the statistical section how the net clinical benefit (number of events for 1000 patients) has been assessed.

In order to look at the impact of each study on outcomes, the authors should perform an influence analysis at least for the primary outcome.

It could also be interesting to perform a sensitivity analysis based on the type of platform used (resorbable versus non-resorbable polymer)

6. PLOS authors have the option to publish the peer review history of their article (what does this mean?). If published, this will include your full peer review and any attached files.

Reviewer #1: No

Reviewer #2: No

---

## [Author Response · Author response to Decision Letter 0]

8 May 2023

Please see attached file entitled "Response to Reviewers" for our complete, itemized responses to reviewer comments.

---

## [Decision Letter · Decision Letter 1]

26 Jul 2023

PONE-D-23-06603R1Short Dual Antiplatelet Therapy Duration After Percutaneous Coronary Intervention in High Bleeding Risk Patients: Systematic Review and Meta-analysisPLOS ONE

Dear Dr. Turgeon,

Thank you for submitting your manuscript to PLOS ONE. After careful consideration, we feel that it has merit but does not fully meet PLOS ONE’s publication criteria as it currently stands. Therefore, we invite you to submit a revised version of the manuscript that addresses the points raised during the review process.

Please address each comment individually in the response letter indicating where changes have been made in the manuscript. 

We look forward to receiving your revised manuscript.

Kind regards,

R. Jay Widmer

Academic Editor

PLOS ONE

Journal Requirements:

Additional Editor Comments:

The reviewers were generally favorable and have a few suggestions regarding the patient selection, treatment type, as well as some comments on tables and figures. Please address these individually in the response letter in a timely fashion, and thank you for your patience on this manuscript revision.

Reviewers' comments:

Reviewer's Responses to Questions

**Comments to the Author**

1. If the authors have adequately addressed your comments raised in a previous round of review and you feel that this manuscript is now acceptable for publication, you may indicate that here to bypass the “Comments to the Author” section, enter your conflict of interest statement in the “Confidential to Editor” section, and submit your "Accept" recommendation.

Reviewer #3: (No Response)

Reviewer #4: All comments have been addressed

2. Is the manuscript technically sound, and do the data support the conclusions?

Reviewer #3: No

Reviewer #4: Yes

3. Has the statistical analysis been performed appropriately and rigorously? 

Reviewer #3: I Don't Know

Reviewer #4: Yes

4. Have the authors made all data underlying the findings in their manuscript fully available?

Reviewer #3: Yes

Reviewer #4: Yes

5. Is the manuscript presented in an intelligible fashion and written in standard English?

Reviewer #3: (No Response)

Reviewer #4: Yes

6. Review Comments to the Author

Reviewer #3: The authors performed a systemic review and meta-analysis of short DAPT vs. standard DAPT after PCI in patients with high bleeding risk. The following were my comments on this article:

1. The study population of these studies was not truly the entire group of patients with high bleeding risk after PCI. In TWILIGHT study, the eligible patients were required to be free from bleeding or ischemic events within the first 3 months after PCI. In TICO study, the key exclusion criteria included increased risk of bleeding due to prior traumatic brain injury or brain surgery within the past 6 months, internal bleeding within the past 6 weeks, and anemia. In STOPDAPT-2 study, the patients who experienced ischemic or bleeding complications during hospital stay post-PCI were excluded. In Master DAPT, the exclusion criteria included any active bleeding requiring medical attention (BARC > 2) within the first month after PCI. And the eligible patients were required to be free from adverse cardiovascular events during the first month after the index PCI. However, most bleeding complications after PCI occur within the first month of DAPT after PCI (doi:10.3390/jcm9061657). So these studies and the result of this meta-analysis could not apply to all patients with high-risk bleeding. On the contrary, they only represented some relatively “stable” high-risk patients after PCI. Patients with high bleeding risk could consider shortening the duration of DAPT and switching to P2Y12 inhibitor monotherapy if they do not have any ischemic or bleeding events within the first 1-3 months after PCI. The title and conclusion may mislead the readers.

2. The subgroup analysis for ACS vs CCS in patients with HRB had many questionable issues. First, the compare group in Master DAPT might receive suboptimal DAPT for ACS patients after PCI, because all ACS patients in compare group received only 6-months DAPT. SMART-DATE trial (https://doi.org/10.1016/S0140-6736(18)30493-8) had proved that shortening the duration of standard 12-month DAPT to 6-month DAPT in ACS patients would significantly increase the risk of recurrent ischemic events. Second, they were only two trials in this subgroup analysis. Third, STOPDAPT-2 ACS study had failed to attest noninferiority to standard 12-months of DAPT for the net clinical benefit with a numerical increase in cardiovascular events. Fourth, there was a similar observation in TWILIGHT-HBR subgroup analysis. The rate of all-cause death, MI, or stroke was increased in ticagrelor monotherapy (6.5% vs. 5.6%; HR 1.16). When the patient number becomes larger, the number would be statistically significant.

3. The duration of DAPT and the choice of subsequent SAPT(aspirin or specific P2Y12 inhibitors) need to take into discussion together. Since most successful clinical trials were applying P2Y12 inhibitor monotherapy after shortened DAPT, it is not appropriate to mix the results of aspirin monotherapy with P2Y12 inhibitor monotherapy. (Most patients in Master DAPT trial were also receiving P2Y12 inhibitor monotherapy after 1-month DAPT)

4. Since this article is a “systemic review and meta-analysis”, I expect the paragraph of “discussion” and “limitation” should be more words and discussion. The gap between current evidence from previous studies and clinical practice could be considered to add into your discussion.

5. In Table 1. “STOPDAPT-2 Total cohort-HBR” and “STOPDAPT-2 HBR” were presented as separate trials. Does ”STOPDAPT-2 total cohort (STOPDAPT-2 + STOPDAPT-2 ACS)-HBR” include the patients enrolled in “STOPDAPT-2 HBR”? In my understanding, “STOPDAPT-2 Total cohort-HBR” and “STOPDAPT-2 HBR” should be viewed as ONE trial.

6. In Figure 2C, It would be better if the order of each included study could be the same as the order used in most of your analyses. So the “TICO-HBR” should be the first one, and “STOPDAPT-2” be the second one.

Reviewer #4: This is a revised submission on short DAPT duration after PCI. Reviewer comments have been adequately addressed.

7. PLOS authors have the option to publish the peer review history of their article (what does this mean?). If published, this will include your full peer review and any attached files.

Reviewer #3: **Yes: **Wen-Han Feng

Reviewer #4: No

---

## [Author Response · Author response to Decision Letter 1]

27 Jul 2023

We thank the PLOS ONE editors and reviewers for their thoughtful review. Below, we have itemized and addressed each comment and refer to the section within the manuscript that has been revised to address each comment.

Reviewer #3: The authors performed a systemic review and meta-analysis of short DAPT vs. standard DAPT after PCI in patients with high bleeding risk. The following were my comments on this article:

1. The study population of these studies was not truly the entire group of patients with high bleeding risk after PCI. In TWILIGHT study, the eligible patients were required to be free from bleeding or ischemic events within the first 3 months after PCI. In TICO study, the key exclusion criteria included increased risk of bleeding due to prior traumatic brain injury or brain surgery within the past 6 months, internal bleeding within the past 6 weeks, and anemia. In STOPDAPT-2 study, the patients who experienced ischemic or bleeding complications during hospital stay post-PCI were excluded. In Master DAPT, the exclusion criteria included any active bleeding requiring medical attention (BARC > 2) within the first month after PCI. And the eligible patients were required to be free from adverse cardiovascular events during the first month after the index PCI. However, most bleeding complications after PCI occur within the first month of DAPT after PCI (doi:10.3390/jcm9061657). So these studies and the result of this meta-analysis could not apply to all patients with high-risk bleeding. On the contrary, they only represented some relatively “stable” high-risk patients after PCI. Patients with high bleeding risk could consider shortening the duration of DAPT and switching to P2Y12 inhibitor monotherapy if they do not have any ischemic or bleeding events within the first 1-3 months after PCI. The title and conclusion may mislead the readers.

We thank the reviewer for this comment. We agree with them that this study does not apply to patients who have actively bled or experienced a subsequent ischemic event during the course of receiving DAPT; however, those patients represent 2 distinct patient groups/scenarios outside the scope of this study. We therefore respectfully disagree that the title or conclusion would mislead readers, particularly as we have used accepted definitions of high bleeding risk (HBR), the primary patient population of this trial, for eligibility into this study (Academic Research Consortium HBR definition and analogous/modified versions of this definition).

2. The subgroup analysis for ACS vs CCS in patients with HRB had many questionable issues. First, the compare group in Master DAPT might receive suboptimal DAPT for ACS patients after PCI, because all ACS patients in compare group received only 6-months DAPT. SMART-DATE trial (https://doi.org/10.1016/S0140-6736(18)30493-8) had proved that shortening the duration of standard 12-month DAPT to 6-month DAPT in ACS patients would significantly increase the risk of recurrent ischemic events. Second, they were only two trials in this subgroup analysis. Third, STOPDAPT-2 ACS study had failed to attest noninferiority to standard 12-months of DAPT for the net clinical benefit with a numerical increase in cardiovascular events. Fourth, there was a similar observation in TWILIGHT-HBR subgroup analysis. The rate of all-cause death, MI, or stroke was increased in ticagrelor monotherapy (6.5% vs. 5.6%; HR 1.16). When the patient number becomes larger, the number would be statistically significant.

We agree with the reviewer that the ACS vs non-ACS subgroup analysis is limited by being reported for only a subset of the studies; however, the main finding remains that there is no heterogeneity of treatment effect explained by this subgroup. Addressing the specific points raised by the reviewer:

• In ACS patients, MASTER DAPT required a minimum of 6 months of DAPT in the control arm; however, most patients received a duration closer to 12 months if they were not receiving an oral anticoagulant at baseline (in which case they would have received dual-pathway therapy). The following figure is from the MASTER DAPT complex PCI subgroup (in which ACS was categorized). Dark blue indicates patients receiving DAPT. We further note that both current American and European guidelines advocate for a reduced DAPT duration of 6 months in ACS patients treated with PCI in HBR patients (prior to the emergence of this evidence). Therefore, we disagree that the comparator arm of MASTER DAPT was suboptimal.

o 

• We acknowledge the results of SMART-DATE; however, we note that other studies comparing 6 to 12 months of DAPT after PCI in non-HBR patients did not find this difference in ischemic events (ref 28 from our manuscript). In either case, these results are not directly applicable to the HBR patients who are the target population of our study.

• The authors note that non-significant differences within individual studies (“trending” toward harm with shorter DAPT) would be statistically significant with a larger sample size, though this did not emerge in our meta-analyses and unlikely based on theoretical and simulation grounds (https://www.bmj.com/content/bmj/348/bmj.g2215.full.pdf).

3. The duration of DAPT and the choice of subsequent SAPT(aspirin or specific P2Y12 inhibitors) need to take into discussion together. Since most successful clinical trials were applying P2Y12 inhibitor monotherapy after shortened DAPT, it is not appropriate to mix the results of aspirin monotherapy with P2Y12 inhibitor monotherapy. (Most patients in Master DAPT trial were also receiving P2Y12 inhibitor monotherapy after 1-month DAPT)

We agree that discussion of which antiplatelet should be retained following DAPT is important, and addressed this in sensitivity analyses (S4 Fig). However, MASTER DAPT was the only trial that allowed ASA monotherapy, and available manuscripts did not disaggregate ASA from P2Y12i use, so this trial was classified as “mixed”.

We discuss this in paragraph 3 of the discussion: “It remains unclear which antiplatelet agent would be best for SAPT following short DAPT. Most patients in the present study received a P2Y12 inhibitor (clopidogrel or ticagrelor) following short DAPT. Additional studies suggest that a P2Y12 inhibitor may be superior to acetylsalicylic acid monotherapy following DAPT. The HOST-EXAM trial conducted in South Korea found that clopidogrel 75 mg daily reduced the risk of MACE and major bleeding compared with ASA 100 mg daily after 6-18 months of DAPT in DES PCI.[35] Additionally, the CAPRIE trial had previously demonstrated lower risk of MI and bleeding with clopidogrel 75 mg daily compared with ASA 325 mg daily.[36]”

In addition to these existing elements, we have added the following to the first (summary) paragraph of the discussion, lines 213-214: “Finally, these results were consistent regardless of clinical presentation, concomitant indication for OAC, or choice of SAPT following DAPT, though most patients received P2Y12 inhibitor monotherapy.”

4. Since this article is a “systemic review and meta-analysis”, I expect the paragraph of “discussion” and “limitation” should be more words and discussion. The gap between current evidence from previous studies and clinical practice could be considered to add into your discussion.

We thank the reviewer for this comment; however, we note that the discussion addresses differences from existing evidence (both in the overall population of patients requiring DAPT, as well as the subgroup of HBR patients), and contrasts to current practice in paragraphs 2 and 3, respectively, as well as addresses other practical elements (which antiplatelet to use as stepdown after DAPT). We have therefore elected to retain the currently succinct discussion.

5. In Table 1. “STOPDAPT-2 Total cohort-HBR” and “STOPDAPT-2 HBR” were presented as separate trials. Does ”STOPDAPT-2 total cohort (STOPDAPT-2 + STOPDAPT-2 ACS)-HBR” include the patients enrolled in “STOPDAPT-2 HBR”? In my understanding, “STOPDAPT-2 Total cohort-HBR” and “STOPDAPT-2 HBR” should be viewed as ONE trial.

We thank the reviewer for the opportunity to clarify. STOPDAPT-2 and STOPDAPT-2 ACS were two distinct trials, with HBR data reported in two separate publications: The STOPDAPT-2 Total cohort-HBR manuscript (both trials) and the STOPDAPT-2 HBR manuscript (only STOPDAPT-2). Although there is overlap in the patient populations between these two manuscripts, we included both within our study (and reported the two substudies separately in Table 1), as they present distinct results, including reporting on different subsets of our outcomes of interest. We indicate in the text and forest plots for each outcome whether the outcomes were reported in both STOPDAPT studies (total cohort-HBR) or the STOPDAPT-2-HBR manuscript (e.g. “All trials except STOPDAPT-2 ACS reported on all-cause death and stent thrombosis”).

To further clarify this point,

• We have added the following footnote to Table 1: “*STOPDAPT-2 Total Cohort-HBR included pooled HBR subgroup data from two distinct trials: STOPDAPT-2 and STOPDAPT-2 ACS, whereas STOPDAPT-2-HBR included HBR subgroup data for STOPDAPT-2 only (i.e. not STOPDAPT-2 ACS).”

• We have revised the following sentence in the Results, lines 136-139: “The included studies (and subgroup analyses) were: MASTER DAPT (the only RCT exclusively enrolling patients with HBR), [19–22] and the HBR subgroup of TWILIGHT[23], TICO[24], and STOPDAPT-2 Total Cohort (which consisted of two trials: STOPDAPT-2 and STOPDAPT-2 ACS trials)[25–27]”

6. In Figure 2C, It would be better if the order of each included study could be the same as the order used in most of your analyses. So the “TICO-HBR” should be the first one, and “STOPDAPT-2” be the second one.

Please note that for all forest plots in the present manuscript, including Figure 2c, trials are deliberately ordered by statistical weight (ascending) to illustrate the relationship between the pooled effect size and the largest trial. We have retained this in the present revision.

---

## [Decision Letter · Decision Letter 2]

22 Aug 2023

Short Dual Antiplatelet Therapy Duration After Percutaneous Coronary Intervention in High Bleeding Risk Patients: Systematic Review and Meta-analysis

PONE-D-23-06603R2

Dear Dr. Turgeon,

We’re pleased to inform you that your manuscript has been judged scientifically suitable for publication and will be formally accepted for publication once it meets all outstanding technical requirements.

Kind regards,

R. Jay Widmer

Academic Editor

PLOS ONE

Additional Editor Comments (optional):

Reviewers' comments:

Reviewer's Responses to Questions

**Comments to the Author**

1. If the authors have adequately addressed your comments raised in a previous round of review and you feel that this manuscript is now acceptable for publication, you may indicate that here to bypass the “Comments to the Author” section, enter your conflict of interest statement in the “Confidential to Editor” section, and submit your "Accept" recommendation.

Reviewer #3: All comments have been addressed

Reviewer #4: All comments have been addressed

2. Is the manuscript technically sound, and do the data support the conclusions?

Reviewer #3: Yes

Reviewer #4: Yes

3. Has the statistical analysis been performed appropriately and rigorously? 

Reviewer #3: Yes

Reviewer #4: Yes

4. Have the authors made all data underlying the findings in their manuscript fully available?

Reviewer #3: Yes

Reviewer #4: Yes

5. Is the manuscript presented in an intelligible fashion and written in standard English?

Reviewer #3: Yes

Reviewer #4: Yes

6. Review Comments to the Author

Reviewer #3: (No Response)

Reviewer #4: The revision and rebuttal to the reviewers comments have been reviewed.

Comments have been adequately responded to in the revision.

7. PLOS authors have the option to publish the peer review history of their article (what does this mean?). If published, this will include your full peer review and any attached files.

Reviewer #3: **Yes: **Wen-Han Feng

Reviewer #4: No

---

## [Editor Report · Acceptance letter]

24 Aug 2023

PONE-D-23-06603R2 

Short Dual Antiplatelet Therapy Duration After Percutaneous Coronary Intervention in High Bleeding Risk Patients: Systematic Review and Meta-analysis 

Dear Dr. Turgeon:

I'm pleased to inform you that your manuscript has been deemed suitable for publication in PLOS ONE. Congratulations! Your manuscript is now with our production department. 

Kind regards, 

on behalf of

Dr. R. Jay Widmer 

Academic Editor

PLOS ONE